# Association between Urban Upbringing and Compulsive Internet Use in Japan: A Cross-Sectional, Multilevel Study with Retrospective Recall

**DOI:** 10.3390/ijerph18189890

**Published:** 2021-09-20

**Authors:** Naonori Yasuma, Daisuke Nishi, Kazuhiro Watanabe, Hanako Ishikawa, Hisateru Tachimori, Tadashi Takeshima, Maki Umeda, Norito Kawakami

**Affiliations:** 1Department of Mental Health, Graduate School of Medicine, The University of Tokyo, Bunkyo-ku, Tokyo 113-0033, Japan; nnyy712@gmail.com (N.Y.); d-nishi@m.u-tokyo.ac.jp (D.N.); hanaco18@gmail.com (H.I.); 2Department of Community Mental Health and Law, National Institute of Mental Health, National Center of Neurology and Psychiatry, Kodaira, Tokyo 187-8553, Japan; 3Department of Public Mental Health Research, National Institute of Mental Health, National Center of Neurology and Psychiatry, Kodaira, Tokyo 187-8553, Japan; 4Department of Public Health, Kitasato University School of Medicine, Sagamihara 252-0374, Kanagawa, Japan; kzwatan@kitasato-u.ac.jp; 5Department of Clinical Epidemiology, Translational Medical Center, National Institute of Mental Health, National Center of Neurology and Psychiatry, Kodaira, Tokyo 187-8553, Japan; tachi@ncnp.go.jp; 6Endowed Course for Health System Innovation, Keio University School of Medicine, Shinjuku-ku, Tokyo 160-8582, Japan; 7Kawasaki City Inclusive Rehabilitation Center, Kawasaki 210-0005, Kanagawa, Japan; takeshima.tadashi@gmail.com; 8Research Institute of Nursing Care for People and Community, University of Hyogo, Akashi 673-8588, Hyogo, Japan; maki_umeda@cnas.u-hyogo.ac.jp

**Keywords:** urban upbringing, compulsive internet use, hierarchical model

## Abstract

The purpose of this study was to show the association between urban upbringing and compulsive internet use (CIU). The interview data of the sample (N = 2431) was obtained from the World Mental Health Japan Second Survey and a multilevel model was used to investigate the association. Multiple imputation was also conducted in this study. Growing up in a large city was significantly associated with higher Compulsive Internet Use Scale (CIUS) scores (γ = 1.65, Standard Error (SE) = 0.45) and Mild CIU + Severe CIU (Exp(γ) = 1.44; 95% Confidence Interval (CI) (1.04–2.00)) compared to growing up in a small municipality after adjusting for both sociodemographic characteristics and psychopathology. This study showed a possible association between urban upbringing and CIU. Future studies with longitudinal design are needed to better understand this association.

## 1. Introduction

Compulsive internet use (CIU) is a serious mental health problem [1]. CIU is defined as excessive or uncontrollable preoccupation with Internet use resulting in disability or distress [1]. Epidemiological research has found a prevalence of CIU in Japan of approximately 7.4% with mild addiction and 7.0% with severe addiction [2]. CIU has been reported to cause family and social problems such as school refusal, unemployment, and domestic violence. [3]. In addition, CIU has been shown to be related to several mental health disorders and suicidality [4,5]. Therefore, investigating its determinants and developing preventive measures are truly important.

Urban living was reported to be one of the determinants of CIU [2,6,7]. This association has been shown in various countries and cultures including Greece, China and Japan, based on various questionnaires such as the Internet Addiction Test (IAT) and the Compulsive Internet Use Scale (CIUS). Several possible mechanisms have been proposed to explain the association between urban living and CIU. First, common mental disorders such as depression and anxiety [8], and non-clinical psychological distress [9], are more prevalent in urban areas. These conditions have been associated with CIU [10,11]. Thus, other mental health conditions may mediate the association between urban living and CIU. Second, CIU has been reported to be associated with lower social support [12]. Social support tends to be lower in urban areas [13]. Lack of social support may be another factor underlying the association between urban living and CIU. However, some studies still report an excess prevalence of CIU in urban settings after adjusting for these covariates [2]. Further studies are needed to investigate the reason why CIU is more prevalent in urban settings.

On the other hand, urban upbringing has been a topic of attention in mental health epidemiology research because exposure to urban living during childhood and adolescence, which are key developmental stages, could lead to mental disorders [14]. Previous studies have revealed that growing up in an urban area during childhood and adolescence is associated with schizophrenia [15] and psychotic experiences in adulthood [16]. Urban upbringing has also been associated with common mental disorders, including depression and alcohol and substance abuse among Blacks in the United States [17]. Urban upbringing may be associated with the development of CIU. However, no research has investigated the association between urban upbringing and CIU. To examine whether urban upbringing or current urban residency is associated with CIU, it would be useful to understand the urban dominance in the prevalence of CIU.

The purpose of this study is to investigate the association between urban upbringing recalled retrospectively and the prevalence of CIU in a survey of the World Mental Health Japan Second Survey (WMHJ2), adjusting for common mental disorders in the past 12 months, psychological distress, social support, and demographic characteristics. In this study, since we now know there is an association between urbanity and CIU [2], in order to investigate the underlying mechanisms further, we plan this study to determine the role of urban upbringing on the association between urbanity and CIU.

## 2. Materials and Methods

### 2.1. Sample

A two-stage random sampling method was used to select the sample for the WMHJ2 Survey from 2013 to 2015 [18]. We divided the survey area into Kanto district (Tokyo Metropolitan area and surrounding areas), Tokai, Hokuriku, Koshin’etsu, Tohoku, Hokkaido (eastern Japan except Kanto) and Kinki, Chugoku, Shikoku, Kyushu, and Okinawa (the western part of Japan). Approximately 5000 samples between the ages of 20 and 75 were selected from community residents in 129 municipalities. The number of samples and average response rate were 2450 and 43.4%, respectively.

### 2.2. Procedure

Trained interviewers visited respondents’ homes and provided them with a full explanation of the study. We gained written informed consent from all respondents. Only respondents who voluntarily agreed to participate in the study were included. Data were collected in two ways: face-to-face interviews and self-administered questionnaires. The World Health Organization Integrated International Diagnostic Interview (WHO-CIDI, version 3.0) was used for the interview survey. When the survey was conducted, no version of WHO-CIDI was available to diagnose the Diagnostic and Statistical Manual of Mental Disorders, Fifth Edition (DSM-5) disorders [19]. Thus, we employed the WHO-CIDI 3.0 based on the Diagnostic and Statistical Manual of Mental Disorders, Fourth Edition (DSM-IV). The authors assert that all procedures contributing to this work comply with the ethical standards of the relevant national and institutional committees on human experimentation and with the Helsinki Declaration. The Research Ethics Committee of The University of Tokyo approved the study (numbers: 10131-(1),-(2),-(3),-(4)).

### 2.3. Measures

#### 2.3.1. Urbanicity

Urbanicity was divided into large cities (Tokyo’s 23 wards, government-ordinance-designated cities), medium cities (over 100,000 people), and small municipalities (other regions). The response rate was 43.5% in large cities, 44.1% in medium cities, and 56.3% in small municipalities.

#### 2.3.2. Urban Upbringing

Urban upbringing was measured using the following question: Were you raised mostly in a large city, medium city, small municipality, a town or a village? The urban upbringing variable was collapsed into three categories: large cities, medium cities, small municipalities. We considered towns or villages as small municipalities. Individuals who grew up in foreign countries or moved around a lot during childhood were excluded.

#### 2.3.3. Compulsive Internet Use

Compulsive Internet Use was measured by the Compulsive Internet Use Scale (CIUS), which consists of 14 items rated on a five-point Likert scale [20]. The severity of CIU is defined as follows: 0 to 17 (no dependence), 18 to 22 (mild), and 23 to 56 (severe). The reliability and validity of the Japanese version of the CIUS has been verified [21]. The internal rate of reliability in the present sample was high (Cronbach’s α = 0.93).

### 2.4. Other Covariates

#### 2.4.1. Common Mental Disorder in the Past 12 Months

Common mental disorder in the past 12 months was measured by the WHO-CIDI 3.0 [22,23] based on the DSM-IV: major depressive disorder, bipolar I and II disorders, dysthymia, panic disorder, generalized anxiety disorder, agoraphobia, social phobia, post-traumatic stress disorder, alcohol abuse and dependence, drug abuse and dependence. Respondents who had at least one mental disorder in the last 12 months was regarded as someone who had experienced a common mental disorder in the past 12 months.

#### 2.4.2. Psychological Distress

Psychological distress was measured by K6 [24], which consists of six items rated on a five-point Likert scale (score range: 0 to 24). Higher scores represent higher degrees of non-clinical depression and anxiety disorders. The Japanese version of K6 has also been verified in terms of its reliability and validity [25]. The internal rate of reliability for the present sample was high (Cronbach’s α = 0.88).

#### 2.4.3. Social Support

Social Support was measured by the Lubben Social Network Scale (LSNS-6) [26], which consists of six items scored on a five-point Likert scale (score range: 0 to 30). The higher the score, the greater the social support. A score less than 12 is defined as indicating social isolation. The Japanese version of LSNS-6 has also been confirmed in terms of its reliability and validity [27]. The internal rate of reliability for the present sample was high (Cronbach’s α = 0.83).

#### 2.4.4. Socio-Demographic Characteristics

We measured the socio-demographic characteristics of age, gender, education, income, and employment. Education was divided into four categories: junior high school graduates, high school graduates, some college, and university graduates or higher. We defined annual household income as the participants’ own earned income, spouse’s income, income from others, social security income, and other income. We imputed missing responses of annual household income by estimating their age, gender, education, employment status and household size [28]. Employment was divided into whether you are currently working or not.

### 2.5. Statistical Analysis

We used a multilevel linear regression analysis to examine the area-level association between urban upbringing and CIU, adjusting for mental disorder in the past 12 months, psychological distress, social support, the type of area currently lived in, and sociodemographic covariates. Psychological distress and social support were considered at both individual and area (city mean score) levels. Other covariates were used at the individual level only. Continuous variables at the individual level (age, psychological distress, and social support) were area mean centered, while continuous scores at area level (psychological distress and social support mean scores) were grand mean centered.

A multilevel logistic regression analysis was also conducted using the severity of CIU (Non-CIU, Mild CIU, and Severe CIU). Non-CIU vs. Mild CIU + Severe CIU and Non-CIU + Mild CIU vs. Severe CIU were compared after adjusting for area-level and individual level covariates.

In these analyses, we conducted a multiple imputation by using the Markov chain Monte Carlo (MCMC) method and predictive mean matching (PMM) to impute missing values. We adjusted for the CIUS scores, urban upbring, the type of area currently lived in, age, gender, education, income, employment, psychological distress, social support, and mental disorder in the past 12 months in the analysis. The number of iterations was 50. For 2450 participants, 19 participants were excluded from the analysis due to lack of area ID. Therefore, 2431 participants were included in the analysis. SPSS (Windows version 27, IBM Corp, New York, NY, USA) was used for statistical analysis. A *p*-value less than 0.05 was considered statistically significant.

## 3. Results

### 3.1. Demographic Characteristics and Prevalence of CIU by Urban Upbringing

Table 1 shows the demographic characteristics and prevalence of CIU by urban upbringing. The prevalence of mild or severe CIU in those who were brought up in large cities was 19.3% (8.8% for mild CIU and 10.5% for severe CIU), in middle cities was 15.7% (8.8% for mild CIU and 6.9% for severe CIU), and in small municipalities was 10.1% (5.3% for mild CIU and 4.8% for severe CIU). The CIUS score distribution was skewed toward the left. The larger the city, the higher the CIUS scores.

### 3.2. Urban Upbringing and CIUS Scores

Table 2 shows the association between urban upbringing and CIUS scores. Growing up in a large city had a significant association with higher CIUS scores compared to growing up in a small municipality (γ = 1.65, Standard Error (SE) = 0.45), while current urban living had no significant association with CIUS scores (γ = 0.75, SE = 0.53).

### 3.3. Urban Upbringing and Mild CIU + Severe CIU and Severe CIU Only

Table 3 shows the association between urban upbringing and mild CIU + severe CIU, and severe CIU only. Growing up in a large city had a significant association with having mild CIU or severe CIU as compared to non-CIU (Exp(γ) = 1.44; 95% CI (1.04–2.00)). However, growing up in a large city was not significantly associated with having severe CIU as compared to non-CIU or mild CIU (Exp(γ) = 1.38; 95% CI (0.93–2.05)). Current urban living was not significantly associated with either mild CIU + severe CIU or severe CIU only (Exp(γ) = 1.02; 95% CI (0.74–1.42)) (Exp(γ) = 1.03; 95% CI (0.69–1.52)).

## 4. Discussion

### 4.1. Main Findings

This is the first study to examine the association between urban upbringing and CIU. Growing up in a large city was significantly associated with higher CIUS scores and mild CIU + severe CIU compared to growing up in a small municipality when adjusting for both sociodemographic characteristics and psychopathology.

### 4.2. Comparison to Previous Findings

Previous studies have reported that urban living was associated with CIU [2,6,7]. Our previous publication also reported the same pattern: that living in a large city was associated with CIU, using the same data from the WMHJ2 [2]. However, in the present analysis, urban upbringing was associated with CIU more strongly than urban living was; urban living was non-significantly and only weakly associated with CIU after adjusting for urban upbringing. Urban upbringing may be a more important determinant of CIU than current urban living.

### 4.3. Potential Mechanisms

Several possible mechanisms could cause the association between urban upbringing and CIU. First, previous studies have consistently shown that growing up in a large city is associated with mental disorders in adulthood, including schizophrenia, psychotic experiences, depression, and alcohol and substance abuse [15,16,17,18], which are also known to be associated with CIU [10,11]. Other mental disorders in early life may facilitate the development of CIU in an urban setting. Second, urban living in childhood and adolescence may provide a greater chance of using the internet in early life because internet access may be better and more affordable in a large city than in rural areas. Furthermore, peer pressure may influence children and adolescents to use the internet in a large city where its use may be more popular than in rural areas [29]. Early life exposure to the internet, and to activities such as SNS and internet gaming, may have conditioned people brought up in a large city to develop CIU earlier or to be prone to CIU in later life. Third, previous studies have shown that exposure during key developmental stages to aspects of poor urban environments, such as air pollution, less green space, and high crime rates, might change the brain structure and gray matter volume [30,31], which could damage the regulation of amygdala activity and stress processing [32]. Moreover, stress associated with urban upbringing might dysregulate the hypothalamic-pituitary adrenal (HPA) axis, which could increase cortisol response and cause acute stress [33]. Through these biological mechanisms, people who grew up in urban areas could have been vulnerable to stress and developed maladaptive coping for stress, e.g., high internet use, which could increase the risk of CIU.

### 4.4. Limitations

There were many limitations to this study. Thus, caution should be used in the interpretation of the results. First, selection bias may have happened because people who have CIU are usually unwilling to participate in the study, which may underestimate the association. Second, information bias could occur when individuals who grew up in large cities were more aware of difficulties related to CIU, making them more willing or hesitant to address such issues in the study. Third, the average age of the subjects was 49 years in this study, which was almost 30 years after childhood and adolescence. While the previous study showed that the effects of urbanicity exposure on mental disorders were more critical during childhood and adolescence, it is possible that other confounding factors may have influenced CIU [14]. For example, socioeconomic status (SES) during childhood and adolescence, which was not assessed in the study, could describe the association between urban upbringing and CIU. This is because children and adolescents with low SES were reported to develop CIU more often than their peers with high SES [34]. In addition, internet access or connection speed might have been other covariates to explain the association between urban upbringing and CIU. Furthermore, chronic diseases such as psoriasis, which causes social impairment and a psychosocial burden might be confounders [35]. Fourth, a type 2 error may have happened because the number of people with severe CIU was a small percentage of the total sample size. Fifth, we defined “urbanicity” in accordance with the official Japanese government city planning. However, other urbanicity indicators may be more accurate to show the association between urban upbringing and CIU. Sixth, the generalization of this study might have been limited due to a lack of measurement variance of CIUS across different cultures. Seventh, the causality between urban upbringing and CIU is still unknown because this study was a cross-sectional study. Finally, there was no version of the WHO-CIDI to diagnose DSM-5 diseases at the time of this study, but if it had been used, the results might have been different. Future studies are needed to reduce these limitations and to allow a comparison with the findings of this study. For example, population-based longitudinal studies from childhood and adolescence that exclude the above biases should be considered.

## 5. Conclusions

While this study had many limitations, as listed above, this study preliminarily showed a possible association between urban upbringing and CIU. Further population-based longitudinal studies in childhood and adolescence are needed to better understand the pathology of this association.

## Figures and Tables

**Table 1 ijerph-18-09890-t001:** Demographics and psychosocial characteristics of the participants (N = 2431).

Type of Area Lived in Growing Up	Large City (N = 466)	Medium City (N = 662)	Small Municipality (N = 1294)	Missing (N = 9)	Overall (N = 2431)
	N (%)	N (%)	N (%)	N (%)	N (%)
CIUS scores ^a^ (Mean)	9.11 (9.8)	8.26 (8.7)	5.60 (8.0)	9.75 (7.4)	7.04 (8.7)
Missing	18 (3.9)	29 (4.4)	85 (6.6)	1 (11.1)	133 (5.5)
Severity of CIU (score range)					
Non CIU ^b^ (0–17)	358 (76.8)	529 (79.9)	1079 (83.4)	6 (66.7)	1972 (81.1)
Mild CIU (18–22)	41 (8.8)	58 (8.8)	68 (5.3)	2 (22.2)	169 (6.9)
Severe CIU (23–56)	49 (10.5)	46 (6.9)	62 (4.8)	0 (0)	157 (6.5)
Missing	18 (3.9)	29 (4.4)	85 (6.6)	1 (11.1)	133 (5.5)
Mental disorder in the past 12 months (yes)	38 (8.2)	37 (5.6)	53 (4.1)	2 (22.2)	130 (5.3)
Missing	0	0	0	0	0
Psychological distress (K6) (Mean)	2.39 (3.3)	2.23 (3.4)	2.02 (3.2)	5.25 (6.3)	2.16 (3.3)
Missing	31 (6.7)	25 (3.8)	98 (7.6)	1 (11.1)	155 (6.4)
Social support (LSNS-6) ^c^ (Mean)	13.54 (5.9)	13.91 (5.7)	13.71 (5.8)	11.13 (5.7)	13.72 (5.8)
Missing	20 (4.3)	17 (2.6)	78 (6.0)	1 (11.1)	116 (4.8)
Type of area lived in currently					
Large City	265 (56.9)	142 (21.5)	259 (20.0)	2 (22.2)	668 (27.5)
Medium City	149 (32.0)	393 (59.4)	416 (32.1)	1 (11.1)	959 (39.4)
Small Municipality	52 (11.2)	127 (19.2)	619 (47.8)	6 (66.7)	804 (33.1)
Missing	0	0	0	0	0
Age (Mean)	49.03 (15.2)	45.42 (14.9)	53.22 (14.5)	55.56 (11.7)	50.30 (15.1)
Missing	0	0	0	0	0
Gender (Male)	233 (50.0)	307 (46.4)	606 (46.8)	5 (55.6)	1151 (47.3)
Missing	0	0	0	0	0
Education					
Junior high school	31 (6.7)	38 (5.7)	149 (11.5)	1 (11.1)	219 (9.0)
High school	147 (31.5)	235 (35.5)	553 (42.7)	2 (22.2)	937 (38.5)
Some college	111 (23.8)	176 (28.7)	325 (25.1)	1 (11.1)	613 (25.2)
University or higher	177 (38.0)	213 (26.6)	264 (20.4)	2 (22.2)	656 (27.0)
Missing	0	0	3 (0.2)	3 (33.3)	6 (0.2)
Income					
≤2.5 million yen	126 (27.0)	195 (29.5)	321 (24.8)	2 (22.2)	644 (26.5)
≤5 million yen	133 (28.5)	183 (27.8)	341 (26.4)	1 (11.1)	658 (27.1)
≤7.5 million yen	101 (21.7)	116 (17.5)	271 (20.9)	3 (33.3)	491 (20.2)
>7.5 million yen	106 (16.7)	168 (25.4)	358 (27.7)	3 (33.3)	635 (26.1)
Missing	0	0	3 (0.2)	0	3 (0.1)
Employment (yes)	264 (56.7)	432 (65.3)	747 (57.7)	5 (55.6)	1448 (59.6)
Missing	0	0	0	2 (22.2)	2 (0.1)

^a^ CIUS: Compulsive Internet Use Scale. ^b^ CIU: Compulsive Internet Use. ^c^ LSNS-6: Lubben Social Network Scale.

**Table 2 ijerph-18-09890-t002:** Urban upbringing and Compulsive Internet Use Scale (CIUS) scores of adult community residents in Japan: multivariate linear mixed model regression (N = 2431).

Urban Upbringing and CIUS Scores	CIUS Scores		
	γ	SE ^a^	*p*
Type of area lived in growing up			
Large City	1.65	0.45	<0.01 ^b^
Medium City	0.68	0.39	0.08
Small Municipality	Reference		
Type of area lived in currently			
Large City	0.75	0.53	0.16
Medium City	−0.28	0.47	0.54
Small Municipality	Reference		

^a^ SE: Standard Error. ^b^ *p* < 0.05.

**Table 3 ijerph-18-09890-t003:** Urban upbringing and mild and severe compulsive internet use (CIU) based on the Compulsive Internet Use Scale (CIUS) scores of adult community residents in Japan: generalized linear mixed model (mixed effects logistic regression) analysis (N = 2431).

Urban Upbringing and Mild and Severe CIU	Non CIU vs. Mild CIU + Severe CIU			Non CIU + Mild CIU vs. Severe CIU		
	Exp(γ)	95% CI	*p*	Exp(γ)	95% CI	*p*
Type of area lived in growing up						
Large City	1.44	1.04–2.00	0.03 ^a^	1.38	0.93–2.05	0.12
Medium City	1.11	0.83–1.49	0.48	1.06	0.73–1.53	0.75
Small Municipality	Reference			Reference		
Type of area lived in currently						
Large City	1.02	0.74–1.42	0.89	1.03	0.69–1.52	0.90
Medium City	0.85	0.62–1.15	0.29	0.79	0.54–1.16	0.23
Small Municipality	Reference			Reference		

^a^ *p* < 0.05.

## Data Availability

The data that support the findings of this study are available on request from the corresponding author. The data are not publicly available due to privacy and ethical restrictions. We did not explain to the participants the public access to the data in the informed consent process. When we receive a reasonable request, the data will be made available after approval by the ethics committee of the University of Tokyo.

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
