# Peer review of "Association between Urban Upbringing and Compulsive Internet Use in Japan: A Cross-Sectional, Multilevel Study with Retrospective Recall"

_ijerph, 2021, doi:10.3390/ijerph18189890_

Round 1

Reviewer 1 Report

In their manuscript, Yasuma et al. reported the outcomes of a study among 2,188 participants regarding the upbringing and the occurrence of Internet addiction. The authors concluded, that an association between urban upbringing and IA is possible. However, in my opinion, there are some limitations within the study, which need to be addressed prior to publication.

  1. If the authors state ORs and 95%-CIs within the results and the abstracts, that’s totally fine and desirable. However, additionally reporting the respective p-value has no beneficial effect and is redundant. Please consider which information is essential in order to keep the manuscript concise.
  2. Citation style is wrong in some places. E.g.: line 49
  3. Some minor spelling errors occurred, e.g.: Tab. 1: Sex – missing bracket.
  4. Considered that part of this data was already presented, the question arises, why the authors excluded the variable ‘upbringing’ within the first analysis. Please explain prior to extensive and detailed revision.
  5. Excluded participants (compared to the previous presentation of this data set; https://doi.org/10.1016/j.psychres.2019.01.094) were n<10 for large and small cities. For middle cities, 300 participants were excluded. Please add this to the discussion of the study results.
  6. Overall, 110 participants were excluded due to single missing variants (<5%). Therefore, I would suggest to use a multiple imputation rather than to merely exclude these patients by default.
  7. The regression models are separate for
  8. I would suggest discussing the potential influence of the respective beauty ideal and chronic, possibly stigmatizing disease like psoriasis on IA.

Author Response

Reply to reviewer: #1

Thank you very much for your valuable and helpful suggestions. We have revised the manuscript in accordance with your suggestions. Revisions are shown in highlighted text in the revised manuscript. We would be pleased if you could have a look at the revised manuscript and check if we responded to your comments appropriately.

Comment 1: If the authors state ORs and 95%-CIs within the results and the abstracts, that’s totally fine and desirable. However, additionally reporting the respective p-value has no beneficial effect and is redundant. Please consider which information is essential in order to keep the manuscript concise.

Reply: Thank you very much for your comment. We excluded description of the p-value from the manuscript.

Comment 2: Citation style is wrong in some places. E.g.: line 49

Reply: Thank you very much for your comment. We revised the wrong citation style.

Comment 3: Some minor spelling errors occurred, e.g.: Tab. 1: Sex – missing bracket.

Reply: Thank you very much for your comment. We revised “sex” into “gender”.

Comment 4: Considered that part of this data was already presented, the question arises, why the authors excluded the variable ‘upbringing’ within the first analysis. Please explain prior to extensive and detailed revision.

Reply: Thank you very much for your comment. In the previous study (Yasuma et al., Psychiatry Res. 2019), we simply intended to clarify the demographic relationship between urbanity and IA; we did not intend to explore the mechanisms underlying the association. Therefore, we excluded the variable ‘urban upbringing’ in the first paper. In this study, since we now know there is an association between urbanity and CIU, in order to investigate the underlying mechanisms further, we plan this study to determine the role of urban upbringing on the association between urbanity and CIU. We added the sentence in the manuscript.

In this study, since we now know there is an association between urbanity and CIU, in order to investigate the underlying mechanisms further, we plan this study to determine the role of urban upbringing on the association between urbanity and CIU. (p2, line:63-66)

Comment 5: Excluded participants (compared to the previous presentation of this data set; https://doi.org/10.1016/j.psychres.2019.01.094) were n<10 for large and small cities. For middle cities, 300 participants were excluded. Please add this to the discussion of the study results.

Reply: Thank you very much for your comment. We were very sorry, but this was just a careless mistake. For middle cities, 378 participants were correct. We revised the Table1.

Comment 6: Overall, 110 participants were excluded due to single missing variants (<5%). Therefore, I would suggest to use a multiple imputation rather than to merely exclude these patients by default.

Reply: Thank you very much for your comment. We reanalyzed the data by using a multiple imputation method in accordance with your suggestion (Table1,2,3). We also added the following sentences in the manuscript.

In these analyses, we conducted a multiple imputation to impute missing responses of psychological distress and social support by estimating their age, gender. Urban upbringing, education, and income were not included in the multiple imputation. (p4, line:156-159)

Comment 7: The regression models are separate for

Reply: The comment is in the middle of a sentence. Therefore, we were very sorry, but we could not answer it.

Comment 8: I would suggest discussing the potential influence of the respective beauty ideal and chronic, possibly stigmatizing disease like psoriasis on IA.

Reply: Thank you very much for giving us your insightful view. We learned a lot from the article as below.

Schielein, M. C., Tizek, L., Schuster, B., Ziehfreund, S., Liebram, C., Eyerich, K., & Zink, A. (2020). Always Online? Internet Addiction and Social Impairment in Psoriasis across Germany. Journal of clinical medicine, 9(6), 1818. https://doi.org/10.3390/jcm9061818

We added psoriasis as one of the possible confounding factors of IA in the manuscript.

Furthermore, chronic diseases such as psoriasis which caused social impairment and psychosocial burden might have been considered as confounders [35]. (p8, line:248-250)

Reviewer 2 Report

Reading the title and abstract of the manuscript promised to be interesting in terms of novelty and findings. Although having some considerations with respect to terminology. However, in reading it I have found certain inconsistencies and important aspects of improvement.

I agree with the authors in one of the statements they make in the manuscript, indicating that more studies are needed to know the reason why CIU is more prevalent in the urban environment. However, I feel that the present manuscript doesn’t contribute in this sense to provide reliable findings that are a scientific advance.

Here are some thoughts, suggestions and ideas for the authors (see attached document).

Author Response

Reply to reviewer: #2

Thank you very much for your valuable and helpful suggestions. We have revised the manuscript in accordance with your suggestions. Revisions are shown in highlighted text in the revised manuscript. We would be pleased if you could have a look at the revised manuscript and check if we responded to your comments appropriately.

Comment 1: Title. It’s recommended not to use the term "internet addiction" as isn’t currently recognized as such by any of the diagnostic manuals (DSM - CIE). It’s suggested to use the term "Compulsive Use of the Internet", since it’s the one used in the instrument with which this variable has been evaluated. On the other hand, the association between urban education and the CIU is proposed, but this must be qualified, since isn’t about present education, but during childhood. Considering that the mean age of the study sample is 48 years, this aspect is relevant. Also, the idea that it’s a cross-sectional study, since it would have to induce a misinterpretation since one of the main study variables is of retrospective type. It’s considered unnecessary to include in the title the survey from which the data comes. It should be indicated in Method.

Reply: Thank you very much for your comment. We revised the title as below in accordance with your suggestion.

Association between Urban Upbringing and Compulsive Internet Use in Japan: A Cross-Sectional, Multilevel Study with retrospective recall. (p1, line:2-4)

Comment 2: Introduction. In the opinion of this reviewer, it’s necessary to review the bibliographic references used in the support of the manuscript. Some of them are irrelevant in relation to the objective and variables analyzed (i.e. Ref 4); however, it’s necessary to expand others to support some considerations that are formulated (i.e. the relationship between CIU and mental disorders or with the habitat typology during upbringing; the association between habitat typology and social support).

Reply: Thank you very much for your comment. We carefully checked the references again and cited them.

Comment 3: Analyzing whether the habitat typology in which the breeding is carried out is associated with CIU seems a very interesting aspect. However, it’s done without considering a series of variables related to habitat typology and over time (retrospective study) that can be determinants to control inherent biases. As it stands, it raises serious questions about the methodology and findings identified. For example, internet access or connection speed are key aspects that vary depending on whether it’s an urban area versus towns or villages. This data is decisive for the development of the CIU. Other variables that could be considered in the literature review and included in the study are the type of employment (if you need internet use), if you are a student, ...

Reply: Thank you very much for your comment. We added the following sentence in the manuscript. We also included the current employment status in the analysis as other confounders.

In addition, internet access or connection speed might have been other covariates to explain the association between urban upbringing and CIU. (p8, line:246-248)

Employment was divided into whether you are currently working or not. (p3, line:136-137)

Socio-demographic variables were compared among urban upbringings. For gen-der, education, income, employment, severity of CIU, mental disorder in the past 12 months and the current level of urban living, the chi-squared test was used. (p3, line:140)

Comment 4: Materials and Methods. Sample. The characteristics of the sample in terms of gender and age aren’t described, as they may also be related to CIU and mental pathology. Age is also key in the use of the internet, both now and in the period of its upbringing. It would be useful to know the response rate according to habitat typology.

Reply: The age range was from 20 to 75 years old, and there were no restrictions based on gender. We added the response rate according to habitat typology.

Approximately 5,000 samples between the ages of 20 and 75 were selected from com-munity residents in 129 municipalities. (p2, line:73)

The response rate was 43.5% in large cities, 44.1% in middle cities, and 56.3% in small municipalities. (p2, line:93-94)

Comment 5: Measures. Municipalities of less than 100,000 inhabitants is very wide. In fact, we could still consider (urban) cities with much lower numbers. Following the question to assess urban education, one should consider "village" and "village". This is essential since social support can be very small in villages where the population is very small and people of the same age range may not exist, contradicting evidence that there is greater social support in the rural area.

Reply: Thank you very much for your comment. As you mentioned, the division of the urbanity was one of the limitations, and interpretation of the results should be cautioned. We added this in the limitation.

Fifth, we defined “urbanicity” in accordance with the official Japanese government city planning. However, other urbanicity indicators may be more accurate to show the association between urban upbringing and CIU. (p8, line:251-254)

Comment 6: What is considered "moving around a lot"? Was any range established? Was it analyzed whether the move went to municipalities with similar characteristics? The question asked did not specify anything about having moved from one municipality to another, how was it asked to establish this exclusion criteria?

Reply: Thank you very much for your comment. "Moving around a lot" is one of the response options to a question on a place lived in childhood. We do not have a further detailed information on this response. Therefore, we have to exclude the respondents with this response from the analysis.

Comment 7: The instruments should be described in more detail. The Cronbach’s index should be reported with this sample for each of the study variables.

Reply: Thank you very much for your comment. We described the instruments more detail and reported the Cronbach’s index.

The internal rate of reliability in the present sample was enough high (Cronbach's α =0.93). (p3, line:105-106)

Psychotic distress was measured by K6 [24], which consists of 6 items rated on a 5-point Likert scale (score range: 0 to 24). Higher scores represent higher degrees of non-clinical depression and anxiety disorders. The Japanese version of K6 have also been verified in its reliability and validity [25]. The internal rate of reliability for the present sample was enough high (Cronbach's α =0.87). (p3, line:118-121)

Social Support was measured by the Lubben Social Network Scale (LSNS-6) [26], which consists of 6 items scored on a 5-point Likert scale (score range: 0 to 30). Higher the score, the greater the social support. A score less than 12 is defined as indicating social isolation. The Japanese version of LSNS-6 have also been confirmed in its reliability and validity [27]. The internal rate of reliability for the present sample was enough high (Cronbach's α =0.83). (p3, line:124-128)

Education was divided into four categories: junior high school graduates, high school graduates, some college, university graduates, or higher. (p3, line:131-132)

Comment 8: Indicate the year of the data collected, as the DSM-5 has been in force since 2013, and for the study use the WHO-CIDI based on the DSM-IV. The authors would have to justify whether or not this affects the results and their interpretation.

Reply: Thank you very much for your comment. We indicated the year of the data collected in the manuscript. When the survey was conducted, no version of WHO-CIDI was available to diagnose DSM-5 disorders. Thus, we employed the WHO-CIDI 3.0 based on the DSM-IV. Diagnoses made by DSM-IV and DSM-5 in the adult population may be slightly different. The findings may be different if we employed DSM-5, while we do not know how different it is. We add this to the limitation.

A two-stage random sampling method was used to select the sample for the WMHJ2 Survey from 2013 to 2015 [18]. (p2, line:70)

When the survey was conducted, no version of WHO-CIDI was available to diagnose DSM-5 disorders[19]. Thus, we employed the WHO-CIDI 3.0 based on the DSM-IV. (p2, line:82-83)

Finally, there was no version of the WHO-CIDI to diagnose DSM-5 diseases at the time of this study, but if it had been used, the results might have been different. (p8, line:257-259)

Comment 9: Results. The analysis and presentation of results reporting on the variables "CIU", "mental disorder in the past 12 months", "psychological distress", "social support" and "income in the home", according to the type of habitat during the upbringing generates serious dudes. The measurement of these variables refers to the present, not to the period of childhood adolescence. It leads to misinterpretation. Too many uncontrolled variables have been able to influence the development of CIU, especially considering that the breeding period could be established up to approximately 18 years, and the sample used has an average of 48 years. In 30 years, multiple changes can happen, both at the social and individual levels, that aren’t evaluated or controlled. Although some of them are indicated in the section Limitations, in my consideration with reviewer, the habitat variable during breeding could have been controlled along with the rest and have considered the variable of habitat typology currently as the variable of comparison. But the latter was already done by the authors in a published article. In my opinion, this is the greatest limitation of the study, since it means that we cannot draw really reliable and explanatory conclusions for a better understanding of the phenomenon

under study. Perhaps selecting a sub-sample closer to the parenting stage (i.e., young adults

aged 20 to 30), will reduce these biases to some extent.

Reply: Thank you very much for your comment. As you mentioned, many other confounding factors were existed. We added the following sentences in the manuscript.

Third, the average age of the subjects was 49 years in this study, which was almost 30 years after childhood and adolescent. While the previous study showed that the effects of urbanicity exposure on mental disorders were more critical during childhood and adolescence, it has been still possible that other confounding factors may have influenced CIU [14]. For example, socioeconomic status (SES) during childhood and adolescence which was not assessed in the study, could describe the association between urban upbringing and CIU. This is because children and adolescents with low SES were reported to develop CIU more often than their peers with high SES [34]. In addition, internet access or connection speed might have been other covariates to explain the association between urban upbringing and CIU. Furthermore, chronic diseases such as psoriasis which caused social impairment and psychosocial burden might have been considered as confounders [35]. (p8, line:239-250)

Comment 10: Discussion. The contribution to scientific knowledge is not relevant and the findings questionable by all mentioned above. The discussion is superficial, it should develop further. A series of explanatory mechanisms are indicated as to why there are more CIU in cities than in the rural environment, but these have not been studied or analyzed in a way that provides new knowledge.

 Reply: Thank you very much for your comment. We strengthened the limitation of this study and added the following sentence in the manuscript.

There were many limitations in this study. Thus, caution should be used in interpretation of the results. (p8, line:234-235)

While this study had many limitations above, this study preliminary showed a possible association between urban upbringing and CIU. Further population-based longitudinal studies in childhood and adolescence should be needed to better understand the pathology of its association. (p8, line:261-264)

Round 2

Reviewer 1 Report

In their manuscript, Yasuma et al. reported the outcomes of a study among 2,188 participants regarding the upbringing and the occurrence of Internet addiction. After the initial revision, the quality of the study was improved tremendously. However, some points remain unclear:

Thank you for performing a multiple data imputation. However, maybe my comment confused the authors. While it might be beneficial to use this method for several analyses (e.g. regressions), descriptive data must not be imputed. Please rather state ‘missing values’ for these numbers and use imputation only where necessary. Additionally, please state all necessary information about your multiple imputation (e.g. numbers of iterations). If you are not sure which values to include, please consider the consultation of a statistician.

Author Response

Reply to reviewer: #1

Thank you very much for your valuable and helpful suggestions. We have revised the manuscript in accordance with your suggestions. Revisions are shown in highlighted text in the revised manuscript. We would be pleased if you could have a look at the revised manuscript and check if we responded to your comments appropriately.

Comment 1: In their manuscript, Yasuma et al. reported the outcomes of a study among 2,188 participants regarding the upbringing and the occurrence of Internet addiction. After the initial revision, the quality of the study was improved tremendously. However, some points remain unclear:

Thank you for performing a multiple data imputation. However, maybe my comment confused the authors. While it might be beneficial to use this method for several analyses (e.g. regressions), descriptive data must not be imputed. Please rather state ‘missing values’ for these numbers and use imputation only where necessary. Additionally, please state all necessary information about your multiple imputation (e.g., numbers of iterations). If you are not sure which values to include, please consider the consultation of a statistician.

Reply: Thank you very much for your comment. We consulted a statistician and reanalyzed the data using multiple imputation. We added the following explanation in the manuscript.

In these analyses, we conducted a multiple imputation by using the Markov chain Monte Carlo (MCMC) and the Predictive Mean Matching (PMM) to impute missing values. We adjusted for the CIUS scores, urban upbring, the type of area currently lived in, age, gender, education, income, employment, psychological distress, social support, and mental disorder in the past 12 months in the analysis. The number of literation was 50 times. For 2450 participants, 19 participants were excluded from the analysis due to lack of the area ID. Therefor, 2431 participants were included in the analysis. SPSS (Windows version 27) was used for statistical analysis. A p-value less than 0.05 was considered statistically significant. (p4, line:152-160)

Reviewer 2 Report

The authors have largely listened to the comments made. In this regard, I congratulate them on their work and efforts. In some respects, the manuscript has improved. The limitations remain important (although they have been listed), which reduces the strength of the findings presented. In any case, they can guide the design of future studies that reduce these limitations and allow a comparison with the findings of this study.

Author Response

Reply to reviewer: #2

Thank you very much for your valuable and helpful suggestions. We have revised the manuscript in accordance with your suggestions. Revisions are shown in highlighted text in the revised manuscript. We would be pleased if you could have a look at the revised manuscript and check if we responded to your comments appropriately.

Comment 1: The authors have largely listened to the comments made. In this regard, I congratulate them on their work and efforts. In some respects, the manuscript has improved. The limitations remain important (although they have been listed), which reduces the strength of the findings presented. In any case, they can guide the design of future studies that reduce these limitations and allow a comparison with the findings of this study.

Reply: Thank you very much for your comment. We described the following sentence in the manuscript.

Future studies should be needed to reduce these limitations and to allow a comparison with the findings of this study. For example, population-based longitudinal studies from childhood and adolescence which excluded above biases should be considered. (p9, line:254-258)

This manuscript is a resubmission of an earlier submission. The following is a list of the peer review reports and author responses from that submission.